# Knowledge of Low Back Pain among Primary School Teachers

**DOI:** 10.3390/ijerph182111306

**Published:** 2021-10-28

**Authors:** Josep Vidal-Conti, Gemma Carbonell, Jaume Cantallops, Pere A Borràs

**Affiliations:** Activity and Sports Research Group (GICAFE), University of the Balearic Islands, 07122 Palma, Spain; gemma.carbonell.b@gmail.com (G.C.); jaume.cantallops@uib.es (J.C.); pa-borras@uib.es (P.A.B.)

**Keywords:** low back pain, knowledge, primary school, teachers

## Abstract

Low back pain (LBP) is a prevalent musculoskeletal disease that affects a large percentage of the working population, including teachers. The World Health Organization has identified the school as an effective environment for improving child health. For this reason, the figure of the teacher is a fundamental piece in the process of knowledge acquisition about postural education and prevention of LBP among schoolchildren. The present study aims to determine the knowledge of postural education and back pain prevention among primary school teachers. This cross-sectional study evaluated 85 primary school teachers from Majorca (Spain), of whom 17.6% were physical education teachers and 82.4% were classroom teachers. The study was based on two different structured and self-administered questionnaires to investigate into specific knowledge about LBP: Low Back Pain Knowledge Questionnaire (LKQ) and COSACUES-AEF Questionnaire. The results demonstrated a lifetime prevalence of LBP of 96.5% with significant differences determined by sex. The knowledge of participants about LBP was 17.3 in LKQ (range scale 0–24) and 4.3 in COSACUES (range scale 1–10). In conclusion, the teachers knowledge is insufficient to carry out an efficient and useful health promotion program among schoolchildren to prevent LBP.

## 1. Introduction

Many musculoskeletal diseases are major health issues that cause disability and have a substantial influence on the general population’s quality of life [1,2]. Low back pain (LBP) is one of the most prevalent musculoskeletal diseases that affects the working population, including teachers [1,3], and is a leading cause of disability in both developed and developing countries [4].

A systematic review published in 2011 clearly suggested that teachers are at risk for developing musculoskeletal disorders [2]. School teachers represent an occupational group among which there appears to be a high prevalence of low back pain (LBP). Examples of this are a 1-month prevalence of LBP of 59.2% in Hong Kong [5], a 12-month prevalence of 45.6% in China [4], 60% in Brazil [6], 64.9% in Kenya [7], and a life-time prevalence of 41.1% in Brazil [8], 44.9% in Turkey [9], 34.8% in France [10], 40.4% and 48.1% in Malaysia [3,11].

On the other hand, LPB among schoolchildren is widely demonstrated, with a lifetime prevalence of LBP in children and teenagers that varies between 3% and 63% [12]. LBP often begins during childhood, however, during adolescence, the prevalence reaches similar values as in adults [13].

The schools are considered a privileged framework for developing an efficient healthcare education program, being a place where children spend most of their time in constant interaction with their peer group. Therefore, the World Health Organization [14] has identified it as an effective environment for improving child health.

For this reason, the figure of the teacher is a fundamental piece in the process of acquiring knowledge about postural education and, concretely, adequate postural habits, to prevent LBP [15]. Postural education is a fundamental pillar on which adequate physical activity and healthy habits are based; this should be developed by physical education teachers [16]. Additionally, in the process of detecting any postural disorder in children, it would be beneficial if, in addition to the doctor being involved, the physical education teacher and the other teachers were also involved [17].

Many postural education programs among children were demonstrated to be effective [18,19], but, in most of them, the intervention was carried out by a researcher, not a teacher. It is important to highlight this fact because, once research is completed, the intervention does not last over time and, consequently, the effects tend to disappear. So, the question is, do teachers have enough knowledge about postural education and how to promote LBP prevention among schoolchildren?

Currently, there are no studies that analyze the knowledge of teachers in relation to postural education. Just one intervention study includes variables about knowledge [20]. This study among teachers with chronic LBP investigated the teachers’ education level based on Alexandre technique lessons combined with an integrative model of behavioral prediction in a three-month follow-up after the intervention. In comparison to the control group, the results revealed that the intervention group’s teacher educational plan facilitated the adoption of Alexandre technique behaviors in teachers and fostered skills and abilities, indirect subjective norms, direct and indirect attitude, direct and indirect perceived behavioral control, and perceived risk.

For all these reasons, it is considered necessary to promote the initial and permanent training of teachers and, specifically, of physical education teachers on postural education in order to be able to promote it appropriately at school age [16].

Therefore, the aim of the present study is to determine the knowledge of postural education and back pain prevention among primary school teachers.

## 2. Materials and Methods

### 2.1. Participants

This cross-sectional study evaluated 85 primary school teachers from Majorca (Spain) Data collection was carried out between February and March 2021. The sample was selected from 10 different clusters (schools) using convenience sampling. All schools received a letter inviting them to participate in the study and informing them about the characteristics and objectives of the study.

### 2.2. Instruments

The study was based on two different structured and self-administered questionnaires to investigate the specific knowledge about LBP: Low Back Pain Knowledge Questionnaire (LKQ) [21] and COSACUES-AEF Questionnaire [22].

The LKQ consists of 16 multiple-choice questions and was used to assess specific knowledge about LBP. The questionnaire covered areas such as the general information about the anatomy of the spine, basic knowledge about LBP, etiology of LBP, classification of LBP, diagnosis, and general management of LBP. The score of items 1–8 is one, and the score of items 9–16 is two. The overall maximum score possible is 24, and general score was classified into one of three levels: low (0–14), moderate (15–18), high (19–24).

COSACUES-AEF consists of 13 multiple-choice questions with three options, where only one is correct. The final score range 1–10 points. The questionnaire aims to measure the knowledge that young people have about health and back care related to the practice of activity and physical exercise. The items refer to knowledge about physical conditioning, muscle strengthening, and stretching or joint mobility. To obtain a final score, the following formula was applied: P = 10 × 1/N(1 × A + B − 1/2 × F), where P is the questionnaire final score, A is the number of correct answers, B is the number of blank answers, F the number of wrong answers, and N is the total number of questions.

Firstly, the participants were asked to fill in their age, kind of teacher (physical education or classroom teacher), prevalence of LBP (lifetime, 1-week and point prevalence). Each participant’s level of physical activity was measured using the short self-administered International Physical Activity Questionnaire (IPAQ) [23]. The questionnaire is based on a recall over the last seven day period. Participants reported the frequency (days per week) and duration (hours) of walking as well as moderate and vigorous physical activity. Responses were converted to Metabolic Equivalent Task minutes per week (MET-min/wk) and then participants were classified into groups (high, moderate, and low physical activity according to the IPAQ scoring protocol [23]) and the methods posted on the IPAQ website (www.ipaq.ki.se, accessed on 26 July 2021).

### 2.3. Procedure

The questionnaire was available on Google Forms, therefore, it was administered at school or home. Teachers were informed about the purpose of the study and its procedure. The study was approved by the Research Ethics Committee of the University of the Balearic Islands (reference number: 130CER19).

### 2.4. Statistical Analysis

After checking for normality with Kolmogorov–Smirnov tests, descriptive characteristics of the sample were calculated, including means with SDs for continuous variables, frequency counts, and percentages for categorical variables.

Differences in all variables, sex (men and women), and kind of teacher (physical education and classroom teacher) were examined with the student’s *t*-test and the chi-squared test for continuous and categorical variables, respectively. To study the association of LBP lifetime prevalence (outcome variable), sex, and kind of teacher (exposure variables), we conducted a binary logistic regression with the calculation of the corresponding odds ratio (OR) and 95% confidence interval (CI). To perform logistic regression, the LBP lifetime prevalence outcome was transformed into a new outcome: those participants who answered “never or only once” were converted into the response “no LBP”, whilst those who answered “sometimes, frequently or almost always” were converted into the response “LBP”.

All statistical analyses were performed using the Statistical Package for Social Sciences (IBM SPSS Statistics for Windows, version 24.0, Armonk, NY, USA) with the significance level set at *p* < 0.05.

## 3. Results

Eighty five respondents completed the questionnaire, of whom 21 were men (24.7%) 64 were women (75.3%), 15 were physical education teachers (17.6%) and 70 were classroom teachers (82.4%).

The results demonstrated a lifetime prevalence of LBP of 96.5%, which means that only 3 out of 85 participants stated that they had never suffered from back pain. Last 7 days prevalence reached 35.3% (n = 30), and point prevalence reached 24.7% (n = 21).

When the participant knowledge about LBP was assessed using LKQ, it was found that the average score of each dimension was 6.52 in general aspects, 3.2 in concepts, and 7.55 in treatments. The total score was 17.27 (over 24), which is the same as saying 7.2 out of 10. When knowledge was assessed using COSACUES questionnaire, the average final score was 4.31 out of 10.

Table 1 shows results by sex. In relation to the prevalence of LBP, chi-squared analysis identified a significant difference between men and women in LBP lifetime prevalence (*p* = 0.01), but not in 7-day prevalence (*p* = 0.601) or point prevalence (*p* = 0.572). In relation to knowledge, no differences were found between men and women either with LKQ or COSACUES questionnaire (*p* > 0.05).

Other characteristics of the LBP among the study population by sex are shown in Table 1.

Type of teacher group (physical education teachers vs. classroom teachers) (Table 2), showed no significant differences in LBP life prevalence (*p* = 0.121), 7-days prevalence (*p* = 0.376) and point prevalence (*p* = 0.338). In relation to the knowledge, in LKQ, no differences were found in total score (*p* = 0.217), but significant differences were found in the dimension of general aspect (*p* = 0.002). Using the COSACUES questionnaire, significant differences were found between physical education teachers and classroom teachers (5.46 and 4.06 over 10 respectively, *p* = 0.011).

Other characteristics of the LBP among the study population by kind of teacher are shown in Table 2.

No significant differences were found between those who have never had LBP and those who have suffered it in the LKQ questionnaire (*p* = 0.341) or COSACUES (*p* = 0.438). Using the LKQ questionnaire, the score was higher for those who have never had LBP (18.38) than those who have suffered it (17.16). On the other hand, using COSACUES, those who have suffered LBP scored higher (4.36) than those who never suffered it (3.79).

Binary logistic regression with LBP lifetime prevalence as a dependent variable and kind of teacher, knowledge (LKQ and COSACUES questionnaires), sex and physical activity as independent variables, showed that the factors independently associated with LBP were sex (OR = 0.06; *p* = 0.011; 95% CI = 0.007–0.526) and knowledge assessed with the COSACUES questionnaire (OR = 1.644; *p* = 0.044; 95% CI = 1.014–2.663) (Table 3).

## 4. Discussion

The present research aimed to determine the knowledge of postural education and back pain prevention among primary school teachers. Furthermore, LBP prevalence and its relationship with postural education knowledge were examined.

The study results showed that lifetime prevalence among teachers was 96.5%, the 7 day prevalence was 35.3%, and point prevalence was 24.7%. Other studies reported a lifetime prevalence of LBP from 34.8 to 48.1% [3,8,9,10,11], and 1 year prevalence from 45.6 to 64.9% [4,6,7]. These differences in the percentages may be due to the strategy for extracting data and the methodology used, sample age, sample size, the definition of LBP, or geographical factors [24]. Despite these differences, most studies show that LBP is a common problem among teachers.

Physical activity was also collected to characterize the sample, and because scientific evidence regarding the role of physical activity in the prevalence of LBP is controversial [25]. Some studies found a curvilinear relationship between them, considering that low and high values of physical activity are associated with an increased risk of back pain [26]. On the other hand, some studies presented different results, as a systematic review that concluded that a high level of physical activity was associated with an increase of LBP [27], or another systematic review that concluded that conflicting evidence was found for the association between physical activity and low back pain in general population [25]. In the present study, the level of physical activity was not associated with LBP.

Regarding the assessment of knowledge of LBP, this is the first study to evaluate the knowledge of LBP in teachers using validated questionnaires. To our knowledge, there is only one study that analyses the knowledge of teachers in relation to postural education, but this study uses a non-validated questionnaire. In that study, 8% of participants reported no knowledge of ergonomics principles, while 72% reported some knowledge, 16% had a reasonable amount of knowledge, and 4% reported extensive knowledge [6].

In our study, two kinds of questionnaires to assess the knowledge of teachers were used. LKQ assesses theoretical aspects, and COSACUES questionnaire assesses practical aspects. The score of LKQ was 17.27 in a 24 point rating scale, 18.27 in physical education teachers and 17.06 in classroom teachers. In either case, we consider that the values are well below what is expected and what is desired. Teachers, who are expected to teach their students, should score close to 24. In comparison, in the validation study of the LKQ questionnaire [21], it was given to 20 healthcare professionals with knowledge on low back pain, who scored an average of 23.55; in another study in nurses, the score was 19.2 [28].

In a study carried out among clinical students using LKQ, 3.5% of participants failed to answer all the questions correctly, in 95.5% less than sixteen questions were answered correctly, and 1.5% answered all the sixteen questions correctly [29]. In another study carried out among nurses, the average score was 19.1 [28], and among Thai adults, the average was 9.2 [30]. In a study carried out among LBP patients attending outpatient physiotherapy treatment in Malawi, only 8.8% of them answered all questions correctly [31]. In any case, despite the fact that there are few studies that evaluate knowledge of postural education, the results obtained should be better. This demonstrates the need to teach postural education from an early age.

In relation to COSACUES questionnaire results, the average final score was 4.31 in a 10 point rating scale, and was identified as an independent risk factor for LBP. Additionally, those who have ever suffered LBP scored higher (4.36) than those who never suffered it (3.79). These results can be explained because people with LBP care to learn about it. In any case, differences were not significant, and both groups’ mean score was low (less than 5). These results are consistent with the findings of other studies that used COSACUES, where participants with LBP had slightly higher scores than those who never suffered it [32,33].

When the results were compared by kind of teacher, significant differences were found between physical education teachers (5.46) and classroom teachers (4.06). These findings may reflect the lack of teacher education (e.g., curriculum of teacher training degrees) in health promotion, specifically in postural education. Thus, it could be that providing information and acquiring knowledge via teacher training degrees, postural education and LBP might have several benefits, such as increasing the knowledge of schoolchildren and their own back care.

To date, there are no specific questionnaires to evaluate the knowledge of postural education among teachers. Due to this, the COSACUES questionnaire was administered, which is prepared for adolescents. For this reason, we consider that the scores obtained are extremely low considering that we are talking about teachers, some of whom are physical education teachers. Two recent studies carried out among secondary education students from Spain, one with participants aged 12 and 13 years, obtained a final score of 2.22 [34], and the other one, with participants aged 12–17 years, scored 2.54 [33]. Both results confirm the low knowledge of postural education that exists in the general population, and more specifically among adolescents.

The implication of these findings could be such that if the participants do not understand what the root cause of LBP is, they cannot reasonably be expected to appreciate the need to teach schoolchildren on how to avoid or manage the pain properly. They might also not be able to appreciate the need for a postural education program for schoolchildren. Different studies have investigated physiotherapy versus educational intervention, and concluded that increasing knowledge via educational intervention was as, and even more, effective than physiotherapy alone in improving pain management and pain resolution [35,36,37].

One limitation of the presented research is the subjectivity of certain results; validated instruments were used to determine LBP, but obtained results are a subjective assessment of the reported LBP. Therefore, results should be interpreted with caution because of their cross-sectional design and the small sample size. Additionally, the COSACUES questionnaire is not validated for the adult population.

A strength of this work is that it is the first cross-sectional study to evaluate the knowledge of postural education in teachers.

It is more advantageous and easier to create healthy behaviors in the youth than it is to try and change already established harmful habits in adults. In this sense, schools play an important role. There have been certain interventions with diverse components that were assessed in randomized trials as possible choices for teaching postural education to elementary school children, with various components adapted to the children’s age range [38]. Once the postural education sessions had been analyzed, all the proposals were adapted to the child population, including active methodology, comic books, games, and characters, among other things, and focused on biomechanics, the spinal column, and posture. As a result, the positive effects on acquiring knowledge and postural habits found in the studies cannot be used to reliably support postural education among schoolchildren. Following this analysis, we believe that intervention efforts should be concentrated on teachers, as they are the most important aspect in a successful intervention in establishing healthy habits.

## 5. Conclusions

This study showed that no differences were found in knowledge related to LBP by sex either with the LKQ or COSACUES questionnaire. However, using the COSACUES questionnaire, significant differences were found between physical education teachers and classroom teachers. Also, knowledge assessed with COSACUES was identified as an independent risk factor for LBP.

In conclusion, our results further strengthen the evidence of the need to enhance knowledge related to LBP and postural education in primary school teachers. Future studies should focus on school interventions, to promote schools as a suitable institution for health promoting.

We agree that school is a perfect place for promoting health among young people. The education of a healthy lifestyle is not an exclusive task of physical education teachers, it is also a task of classroom teachers. However, to be effective and useful, teachers are required to be properly trained in postural education, and this is not the current reality.

## Figures and Tables

**Table 1 ijerph-18-11306-t001:** Characteristics of the total sample by sex.

	Total (n = 85)		Men (n = 21)		Women (n = 64)		*p*
	X	(SD)	X	(SD)	X	(SD)	
LKQ dimension (range scale)							
General Aspects (0–9)	6.52	(1.21)	6.71	(1.19)	6.45	(1.22)	t = 0.856 *p* = 0.394
Concepts (0–4)	3.20	(0.94)	2.86	(1.28)	3.31	(0.77)	t = −1.545 *p* = 0.135
Treatments (0–11)	7.55	(2.19)	7.19	(2.18)	7.67	(2.20)	t = −0.873 *p* = 0.385
Total score (0–24)	17.27	(3.43)	16.76	(3.88)	17.44	(3.28)	t = −0.782 *p* = 0.436
Total score (0–10)	7.20	(1.43)	6.98	(1.62)	7.27	(1.37)	
COSACUES (range scale 1–10)	4.31	(1.95)	4.89	(2.42)	4.12	(1.75)	t = 1.342 *p* = 0.191
METs	3191	(4092)	3998	(3016)	2926	(4376)	t = 1.042 *p* = 0.300
	n	(%)	n	(%)	n	(%)	
LKQ categories							
Low	17	(20)	3	(14.3)	14	(21.9)	X^2^ = 1.834 *p* = 0.400
Moderate	34	(40)	11	(52.4)	23	(35.9)	
High	34	(40)	7	(33.3)	27	(42.4)	
Kind of teacher							
PE teacher	15	(17.6)	7	(33.3)	8	(12.5)	X^2^ = 4.722 *p* = 0.046
Classroom teacher	70	(82.4)	14	(66.7)	56	(87.5)	
LBP prevalence							
Never	3	(3.5)	3	(14.3)	0	(0)	X^2^ = 13.325 *p* = 0.010
Only once	5	(5.9)	2	(9.5)	3	(4.7)	
Sometimes	48	(56.5)	13	(61.9)	35	(54.7)	
Frequently	26	(30.6)	3	(14.3)	23	(35.9)	
Almost always	3	(3.5)	0	(0)	3	(4.7)	
LBP ever (yes)	77	(90.6)	16	(76.2)	61	(95.3)	X^2^ = 6.781 *p* = 0.020
LBP 1 week prevalence (yes)	30	(35.3)	6	(28.6)	24	(37.5)	X^2^ = 0.552 *p* = 0.601
LBP point prevalence (yes)	21	(24.7)	4	(19)	17	(26.6)	X^2^ = 0.480 *p* = 0.572
Phisical Activity Level							
Low	18	(21.2)	5	(23.8)	13	(20.3)	X^2^ = 10.226 *p* = 0.006
Moderate	36	(42.4)	3	(14.3)	33	(51.6)	
High	31	(36.5)	13	(61.9)	18	(28.1)	

**Table 2 ijerph-18-11306-t002:** Characteristics of the sample by kind of teacher.

	Total (n = 85)		PE Teachers (n = 15)		Classroom Teachers (n = 70)		*p*
	X	(SD)	X	(SD)	X	(SD)	
LKQ (range scale)							
General Aspects (0–9)	6.52	(−1.21)	7.13	(0.64)	6.39	(1.27)	t = 3.337 *p* = 0.002
Concepts (0–4)	3.20	(0.94)	3.33	(0.98)	3.17	(0.93)	t = 0.606 *p* = 0.546
Treatments (0–11)	7.55	(2.19)	7.80	(1.82)	7.50	(2.27)	t = 0.479 *p* = 0.633
Total score (0–24)	17.27	(3.43)	18.27	(2.25)	17.06	(3.61)	t = 1.244 *p* = 0.217
Total score (0–10)	7.20	(1.43)	7.61	(0.94)	7.11	(1.50)	
COSACUES (range scale 1–10)	4.31	(1.95)	5.46	(2.54)	4.06	(1.73)	t = 2.596 *p* = 0.011
METs	3191	(4092)	5512	(8038)	2693	(2418)	t = 2.496 *p* = 0.015
	n	(%)	n	(%)	n	(%)	
LKQ categories							
Low	17	(20)	0	(0)	17	(24.3)	X^2^ = 4.655 *p* = 0.098
Moderate	34	(40)	8	(53.3)	26	(37.1)	
High	34	(40)	7	(46.7)	27	(38.6)	
Sex (men)			7	46.7	14	20	X^2^ = 4.722 *p* = 0.046
LBP prevalence							
Never	3	(3.5)	2	(13.3)	1	(1.4)	X^2^ = 7.300 *p* = 0.121
Only once	5	(5.9)	1	(6.7)	4	(5.7)	
Sometimes	48	(56.5)	9	(60)	39	(55.7)	
Frequently	26	(30.6)	2	(13.3)	24	(34.3)	
Almost always	3	(3.5)	1	(6.7)	2	(2.9)	
LBP ever (yes)	77	(90.6)	12	(80)	65	(92.9)	X^2^ = 2.395 *p* = 0.144
LBP 1 week prevalence (yes)	30	(35.3)	7	(46.7)	23	(32.9)	X^2^ = 1.032 *p* = 0.376
LBP point prevalence (yes)	21	(24.7)	2	(13.3)	19	(27.1)	X^2^ = 1.266 *p* = 0.338
Physical Activity Level							
Low	18	(21.2)	2	(13.3)	16	(22.9)	X^2^ = 7.232 *p* = 0.027
Moderate	36	(42.4)	3	(20)	33	(47.1)	
High	31	(36.5)	10	(66.7)	21	(30)	

**Table 3 ijerph-18-11306-t003:** Logistic regression results for determining LBP prevalence.

	OR	*p*	I.C. 95.0%	
Classroom teacher	5.210	0.109	0.691	39.272
LKQ	0.581	0.163	0.271	1.245
COSACUES	1.644	0.044	1.014	2.663
Sex	0.060	0.011	0.007	0.526
Physical Activity				
High Level		0.509		
Moderate level	0.405	0.428	0.043	3.800
Low Level	1.896	0.620	0.151	23.775

## Data Availability

The data are available on request from the Physical Activity and Sports Research Group, University of the Balearic Islands. The request should be formulated and sent to josep.vidal@uib.es.

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
