# Peer review of "Knowledge of Low Back Pain among Primary School Teachers"

_ijerph, 2021, doi:10.3390/ijerph182111306_

Round 1

Reviewer 1 Report

  • The sentence from line 30 to 36 could be organised better. It is too long and hard to follow now.
  • Line 35 "44.9)" I believe it needs to be "44.9%"
  • line 146 says, "Other characteristics of the LBP among the study population by age are shown in Table 2." but table 2 shows characteristics of the sample by kind of teacher.

Reviewer 2 Report

Thank you for submitting your manuscript “Knowledge of low back pain among primary school teachers”. It is interesting that school teachers have limited knowledge of back pain prevention. It would be interesting to investigate this concept further among teachers from different countries and unpack possible cultural differences for back pain prevention in this population group. Below are some comments and feedback for consideration.

Abstract

There are no results are reported in order to support the abstract’s conclusion.

Introduction

Please quantify large statements and include supporting evidence. For example,

 “Musculoskeletal diseases are serious health issues that cause disability…” not all MSK diseases serious. An episode of acute low back pain may not be serious at all.

And other statements are not referenced at all. For example “…. and is a leading cause of disability in both developed and developing countries.”

In the statement, “A systematic review developed in 2011 already clearly suggested that teachers are at risk for developing musculoskeletal disorders”, I think the authors mean “published” not “developed”.

The introduction would be clear if the population was stated. For example,

-Besides, in the process of detecting any postural disorder, it would be beneficial if in addition to the doctor being involved, the physical education teacher and the other teachers were also involved [17]” Is this statement referring to children or teachers?

-“Many postural education programs demonstrated to be effective [18,19], but in ….” Is the program effective in children or in teachers?

-“In comparison to the control group, the results revealed that the intervention group's teacher educational plan facilitated the adoption of Alexandre technique behaviors and fostered skills and abilities, in-direct subjective norms, direct and indirect attitude, direct and indirect perceived behavioral control, and perceived risk” In children?

Methods

How were the schools selected and how was the sample size determined?

What was the purpose of clustering schools when the study is not a cluster randomised trial, only a cross-sectional study?

Details of the participants (e.g. gender, teacher types) are study results and should be presented in the results section, not the methods section.

The COSACUES questionnaire is presented in both upper- and lower-case text (it should be in upper case text throughout). Also, what is the overall scoring of COSACUES-AEF? The scoring is described for the LKQ questionnaire.

What data were collected is not clear. If can be inferred from the statistical analysis but it is not comprehensively reported.

How was missing data handled?

Results

Regarding the statement, “The results demonstrated a lifetime prevalence of LBP of 96.5%, which means that 3 out of 85 participants stated that they had suffered from back pain at least once in their lives.”, do the authors mean that 3 participants did not have LBP?

The statement “Using the dichotomic variable created from the question related to LBP lifetime prevalence, it shows 90.6% of LBP prevalence” contradicts the first sentence in this paragraph.

Table 1. Footnote is missing detailing abbreviations e.g. METs. What does “over” mean. Do the authors mean a score greater than X? There is a negative standard deviation score, is this correct?

Discussion

The discussion, at times, is descriptive of the results and paragraphs are sometimes only 1-2 sentences.

The latter also occurs in the introduction. The minimal use of repeating the results would strengthen and deeper the discussion. The authors could consider what future research is needed considering the impact and change they wish to achieve or initiate. Furthermore, the content of the discussion does not match the conclusion of the abstract. I was expecting a deeper discussion after reading the abstract “Results showed that the knowledge of teachers is too low to carry out an efficient and useful health promotion program among schoolchildren to prevent LBP.”

I am unclear why the COSACUES questionnaire was used if it is not validated (line 198-199).

The conclusion is more a commentary rather than a summary of the study.

Reviewer 3 Report

The article deals with a very important problem of the possibility of teaching students the knowledge of posture and prophylaxis of back pain by primary school teachers. Therefore, I believe it should be published. However, there are some ambiguities in it that should be removed:

  1. The authors adopted the assumption that the knowledge about postural education and back pain prevention can be effectively transferred by teachers who themselves have it at an appropriate level. The level of this knowledge was checked using the LKQ and COSACUES-AEF questionnaires. These questionnaires were described very succinctly. We only learn about the LKQ questionnaire that it contains general, conceptual and treatments aspects, about the COSAQUES questionnaire - that it has a final result. No criteria were given for the evaluation of the obtained results. Nevertheless, the authors state without any justification that "Teachers, who are expected to teach their students, should score (in LKQ questionnaire) close to 24".
  2. In accordance with the rules, tables should be prepared in such a way that they are understandable regardless of the text of the article. All abbreviations used in the tables must be explained. Unfortunately, this is not the case: which means the abbreviation METs in tables 1 and 2, as well as PA. The methods for measuring these parameters are not described in the “Instuments”, there is nothing about them either in part of the “Results” or in the “Discussion”.
  3. The obtained results (especially those statistically significant) should be interpreted. The authors stated that the knowledge assessed on the basis of the COSACUES questionnaire was an independent risk factor for LBP (OR = 1.644; p = 0.044; 95% CI = 1.014-2.663), but they did not say anything about it.

Round 2

Reviewer 2 Report

Thank you for considering my suggestions.